# MicroRNAs, Tristetraprolin Family Members and HuR: A Complex Interplay Controlling Cancer-Related Processes

**DOI:** 10.3390/cancers14143516

**Published:** 2022-07-20

**Authors:** Cyril Sobolewski, Laurent Dubuquoy, Noémie Legrand

**Affiliations:** Inserm, CHU Lille, U1286-INFINITE-Institute for Translational Research in Inflammation, Univ. Lille, F-59000 Lille, France; laurent.dubuquoy@inserm.fr (L.D.); noemie.legrand@univ-lille.fr (N.L.)

**Keywords:** microRNAs, AUBP, HuR, TTP, cancers

## Abstract

**Simple Summary:**

AU-rich Element Binding Proteins (AUBPs) represent important post-transcriptional regulators of gene expression by regulating mRNA decay and/or translation. Importantly, AUBPs can interfere with microRNA-dependent regulation by (*i*) competing with the same binding sites on mRNA targets, (*ii*) sequestering miRNAs, thereby preventing their binding to their specific targets or (*iii*) promoting miRNA-dependent regulation. These data highlight a new paradigm where both miRNA and RNA binding proteins form a complex regulatory network involved in physiological and pathological processes. However, this interplay is still poorly considered, and our current models do not integrate this level of complexity, thus potentially giving misleading interpretations regarding the role of these regulators in human cancers. This review summarizes the current knowledge regarding the crosstalks existing between HuR, tristetraprolin family members and microRNA-dependent regulation.

**Abstract:**

MicroRNAs represent the most characterized post-transcriptional regulators of gene expression. Their altered expression importantly contributes to the development of a wide range of metabolic and inflammatory diseases but also cancers. Accordingly, a myriad of studies has suggested novel therapeutic approaches aiming at inhibiting or restoring the expression of miRNAs in human diseases. However, the influence of other trans-acting factors, such as long-noncoding RNAs or RNA-Binding-Proteins, which compete, interfere, or cooperate with miRNAs-dependent functions, indicate that this regulatory mechanism is much more complex than initially thought, thus questioning the current models considering individuals regulators. In this review, we discuss the interplay existing between miRNAs and the AU-Rich Element Binding Proteins (AUBPs), HuR and tristetraprolin family members (TTP, BRF1 and BRF2), which importantly control the fate of mRNA and whose alterations have also been associated with the development of a wide range of chronic disorders and cancers. Deciphering the interplay between these proteins and miRNAs represents an important challenge to fully characterize the post-transcriptional regulation of pro-tumorigenic processes and design new and efficient therapeutic approaches.

## 1. Introduction

Trans-acting factors controlling the fate of mRNA importantly contribute to gene expression regulation. Among them, intense efforts have been devoted to non-coding RNAs, mostly miRNAs, which mainly bind to the 3′Untranslated Region (3′-UTR) of target mRNAs and promote either mRNA degradation and/or translation inhibition [1]. The importance of miRNAs in physiological and pathological processes is now well-established. For most human diseases, the altered miRnome has been identified through high-throughput approaches and has suggested appealing therapeutic approaches to target deregulated miRNAs [2,3]. However, increasing studies indicate that the post-transcriptional regulation of gene expression is more complex than originally thought and requires other regulators, such as long-non-coding RNAs (lncRNA) or RNA-Binding Proteins (RBPs) [4,5,6]. AU-Rich Element Binding Proteins (AUBPs) are RBPs of particular importance due to their ability to bind to AU-rich sequences in the 3′UTR of immediate response genes (e.g., TNF-α, cyclooxygenase-2) and promote either their decay or translation inhibition [7,8]. Similarly, for miRNAs, deregulated expression/activity of AUBPs strongly contributes to the development of a wide range of diseases due to their ability to control a whole set of genes associated with pathological processes, including inflammation (e.g., Tumor Necrosis Factor alpha, TNFα), fibrosis, metabolism, or carcinogenesis [9]. These proteins can interfere, compete, or synergize with miRNA-dependent regulation [10,11] and conversely, miRNAs can also regulate the expression of AUBPs. However, this complex interplay is poorly considered in most studies (e.g., miR-21, [12]; let-7 [13], tristetraprolin [14]), and thus, our current comprehension of miRNA/RBPs biology in physiological and pathological processes is strongly limited, as evidenced by the increasing number of contradictory findings in the literature regarding microRNA’s functions [15,16]. This can be explained by the difficulty of fully recapitulating this degree of complexity in current preclinical models. In this review, our goal is to describe part of this complex regulatory network in human cancers, with an emphasis on the interplay between the most studied AUBPs (TTP family members and HuR) and miRNAs-dependent regulation.

## 2. MiRNAs: From Biogenesis to a Complex Regulatory Network

MicroRNAs are small endogenous ≈20 nucleotides non-coding sequences promoting mRNA decay or translation inhibition through their ability to bind to a specific seed sequence in the 3′UTR of target mRNAs [17,18,19]. In the canonical model, the perfect complementarity between the mRNA and the seed sequence of miRNAs leads to mRNA decay, while imperfect complementarity is thought to mediate translation inhibition [19]. However, this model is not completely understood. MiRNA biogenesis starts with the generation of a primary miRNA transcript (pri-miRNA) by RNA polymerase II/III [19]. Then, the pri-miRNA is cleaved by the microprocessor complex comprised of the ribonuclease III enzyme, Drosha and the RBP DGCR8 (DiGeorge Syndrome Critical Region 8), thereby producing a precursor miRNA (pre-miRNA), which is exported to the cytosol via the Exportin5/RanGTP [19]. In the cytosol, pre-miRNAs are processed by the RNase III endonuclease Dicer, which removes the terminal loop, thereby producing a mature miRNA duplex comprised of a guide strand and a complementary passenger strand (miRNA*) [19]. Although it was originally thought that only the guide strand is important for targeted gene silencing, it is now clear that depending on the cell type or the context (e.g., inflammation, deregulated metabolism), both strands can associate with argonaute proteins and incorporates into the RNA-Induced Silencing Complex (RISC), where it binds to its target mRNAs [19]. MiRNAs can regulate hundreds of mRNAs, and conversely, one mRNA can be regulated by multiple miRNAs. Moreover, depending on the cellular or physiological context (e.g., nutrients deprivation, hypoxia, treatment with anti-tumor agents), miRNAs may exert different functions and regulate different targets. Finally, miRNAs are subjected to sequence editing during processing (e.g., A-to-I), which strongly alters their binding properties and the choice of the strand incorporated into the RISC [20,21]. Considering this complexity, the current models and approaches to study miRNA-dependent regulation appear quite simplistic. Another layer of complexity comes from other trans-acting factors that control the fate of mRNAs and interfere with miRNA-dependent regulation or are regulated by miRNAs. This includes lncRNAs but also RBPs, such as AUBPs, which can compete with the same binding sites with miRNAs, cooperate with miRNAs or act as “sponges” to sequester miRNAs, thereby preventing their binding to their target mRNA [22,23]. Whether a miRNA is active or not, therefore, depends on the availability of the seed sequence on target mRNA and the bioavailability of miRNAs in each system. However, for most studies, only miRNAs with the most deregulated expression are considered in the pathophysiology of human diseases. This choice is potentially wrong, given that the deregulated expression of mRNAs does not always translate into a deregulated activity [24,25]. Finally, the complexity of such a regulatory network is further enhanced by the ability of several miRNAs to control the expression of AUBPs. Deciphering this complex regulatory network is therefore crucial for identifying bioavailable and active miRNAs involved in physiological and pathological processes. Emerging databases (e.g., SIMIRA) can provide some help in predicting the common targets between RBPs and miRNAs (http://vsicb-simira.helmholtz-muenchen.de, accessed on 1 June 2022). Immunoprecipitation assays of RBPs coupled with RNAseq are also highly relevant for identifying all non-coding RNAs bound to RBPs [26,27,28]. These data are publicly available in specific databases (http://clipdb.ncrnalab.org, accessed on 1 June 2022; http://rbpdb.ccbr.utoronto.ca/, accessed on 1 June 2022). Finally, the complexity of this regulatory network should be better represented in the in vitro and in vivo models. Developing models that consider the interplay between miRNAs and these other trans-acting factors represents a relevant approach to fully characterize the role of these post-transcriptional regulators and uncover novel and efficient therapeutic approaches.

## 3. AUBPs Interfering with miRNAs-Dependent Regulation

AUBPs represent a family of RBPs controlling gene expression at the post-transcriptional level [9,29]. These proteins have a high affinity for Adenine and Uridine-rich elements (ARE: e.g., AUUUA pentamer) within the 3′UTR of several immediate-early response genes (which represent about 5–8% of the human transcriptome) and mediate their decay and/or impair their translation through processes involving their recruitment into small cytoplasmic compartments, namely, Stress Granules (SGs) or processing-bodies (P-bodies), respectively [9,29]. More than 20 different AUBPs have been identified and display varying expression patterns depending on the tissue type or the physiological context (e.g., inflammation, hypoxia, chemotherapy) [9]. While some AUBPs promote mRNA decay (e.g., tristetraprolin family members:), others are mostly stabilizing their target mRNAs (e.g., Human Antigen R, HuR) and/or blocking their translation (e.g., T-cell Intracellular Antigen-1, TIA1) [9,29,30,31,32]. However, emerging studies showing opposite functions currently challenge this very strict and dogmatic view. The activity and subcellular localization of these proteins are regulated by various post-translational modifications (e.g., phosphorylation, methylation), depending on the cellular context (e.g., mitosis, endoplasmic reticulum, and oxidative stress) [33,34,35]. Alteration of AUBPs expression/activity contributes to deregulated expression patterns of genes involved in inflammatory (e.g., *PTGS2*, *TNFA*) [31,36,37], metabolic (e.g., *FGF21*, *PTEN*) [38,39] and carcinogenic processes (oncogenes/tumor suppressors, e.g., *MYC*, *P53*) [40,41] and thus contributes to the development of a wide range of chronic metabolic/inflammatory diseases and cancers [9,29]. Importantly, these proteins can also compete with each other for the same mRNA targets (e.g., HuR and TTP for COX-2) [30], and increasing evidence indicates that these proteins can regulate gene expression by interfering with miRNA-dependent regulation [42,43]. Herein, we discuss these complex interactions and their role in cancer development, with an emphasis on HuR and TTP family members. The different connections between HuR, TTP family members and miRNA-dependent regulation of gene expression are discussed below and are summarized in Figure 1 and Figure 2.

### 3.1. HuR

HuR (encoded by *ELAVL1: Embryonic-Lethal Abnormal Vision in Drosophila*) is ubiquitously expressed AUBPs, which possess two tandems RNA-Recognition motif (RRM), and a hinge region followed by a third RRM [44,45]. The hinge region is subjected to various post-translational modifications (e.g., phosphorylations, methylation) and is mostly involved in the nucleo-cytoplasmic shuttling of HuR [46,47]. In the cytosol, HuR exerts its mRNA stabilizing function by targeting ARE and competing or displacing other destabilizing factors (e.g., miRNAs, TTP family members) with overlapping binding sites [48]. Finally, HuR can also sequester or potentiate miRNA functions [42]. HuR expression is frequently upregulated in human cancers and plays a compelling role in carcinogenic processes [44,45]. HuR has mostly been associated with tumor-promoting functions due to its ability to stabilize a whole set of oncogenic and pro-inflammatory transcripts (e.g., COX-2) [49]. However, increasing evidence indicates that HuR can interfere with miRNA-dependent regulation [42] and can exert tumor suppressive functions. Understanding the molecular traits associated with HuR’s functions is, therefore, determinant prior to any therapeutic intervention, as suggested by HuR inhibitors (e.g., MS-444, DHTS, KH-3) in several cancers [50,51,52].

#### 3.1.1. Competitive Antagonism between HuR and miRNAs

*HuR-miR-21:* MiR-21 is a well-known miRNA frequently upregulated in human cancers and associated with pro-tumorigenic functions and a poor clinical outcome in patients [53,54,55]. Increasing evidence indicates that HuR can prevent the miR-21-dependent function. Indeed, the RBP, lupus antigen (La), together with HuR, binds to the 3′UTR of *PDCD4* (Programmed Cell Death Protein 4), a tumor suppressor, to prevent the binding of miR-21 [56]. Although HuR is mostly considered a tumor-promoting factor, these findings suggest a tumor suppressive role of HuR. The tumor-promoting properties of miR-21 have essentially been uncovered in vitro in transformed cancerous cell lines [53]. However, our previous study demonstrated for the first time that, in contrast to in vitro findings, miR-21 could also exert tumor suppressive properties [57]. Therefore, these data suggest a context-dependent function of HuR and miR-21. In agreement, the binding of HuR to *PDCD4* mRNA is prevented in oxidative stress conditions (H_2_O_2_) due to the phosphorylation of HuR by ERK8 (Extracellular Signal Regulated Kinase 8), as evidenced in Hela cells [58]. Taken together, these data suggest cautions regarding the dogmatic statement of miR-21 functions in cancer cells, based on the miRNA expression only, without considering its activity. A better characterization of this interplay is therefore required prior to the development of any therapeutic approaches aiming at targeting miR-21 or HuR in cancers.

*HuR-miR-26a/b.* In breast cancer, the resistance to tamoxifen treatment is associated with a competition between HuR and miR-26a/b, which regulates Erb-B2 receptor tyrosine kinase 2 (*ERBB2*) [59]. Indeed, in tamoxifen-resistant (TAMR) estrogen receptor-positive (ER+) breast cancer cells, HuR stabilizes *ERBB2* mRNA, while miR-26a/b [59] inhibits its translation. Accordingly, overexpression of miR-26a/b or HuR silencing can restore the sensitivity of cancer cells to tamoxifen.

*HuR-miR-34b-5p.* In colon cancer cells (SW620, HT29, HCT116, LoVo and RKO cells), HuR binds and stabilizes the lncRNA *OIP5-AS1* (OIP5 Antisense RNA 1) [60]. MiR-34b-5p competes with HuR for the binding to *OIP5-AS1*. Moreover, miR-34b-5p directly inhibits the expression of HuR. Accordingly, in CRC (colorectal cancer) patients, both *OIP5-AS1* and HuR expressions are elevated as compared to matched non-tumoral tissues. This interplay importantly promotes carcinogenesis because of the ability of *OIP5-AS* to trigger the PI3K/AKT pathway.

*HuR-**OIP5-AS1 and miR-424.* As described previously in colon cancer, HuR stabilizes the lncRNA *OIP5-AS1* in human cervical cancer cells (Hela cell line). Moreover, *OIP5-AS1* acts as a decoy to prevent the binding of HuR on mRNA targets, thereby reducing cancer cell proliferation. In contrast, miR-424 competes with HuR and downregulates *OIP5-AS1* expression [61].

*HuR-miR-16.* In breast cancer cells, the regulation of cyclin E1 mRNA (*CCNE1*) depends on the interplay between HuR and miR-16 [62]. Indeed, in a luciferase-reporter construct, HuR blocks the binding of miR-16 on the 3′UTR of *CCNE1* in MCF-7 cells. While this effect was not observed with the endogenous *CCNE1* transcript, miR-16 was able to prevent the HuR-dependent *CCNE1* stabilization. Together, these findings highlight the importance of this interplay during cell cycle progression.

*HuR-miR-194/miR-206.* Nucleolin (*NCL*) is overexpressed in several malignancies and promotes cancer cell proliferation [63]. This overexpression of *NCL* is mediated by the elevated expression of HuR and the downregulation of miR-194 and miR-206, which competitively bind to *NCL* 3′UTR, as evidenced in MCF-7 and MDA-231 cells (breast cancer cells) [64].

*HuR-OSER1-AS1-miR-17-5p. OSER1-AS1* is a lncRNA with tumor suppressive properties in Non-Small Cell Lung Cancer (NSCLC). In agreement, the loss of its expression in tumor from patients correlates with a poor clinical outcome. In the H1299 cell line, *OSER1-AS1* binds to HuR and acts as a decoy to prevent HuR from stabilizing its mRNA target. On the other hand, miR-17-5p competes with HuR to bind and downregulate *OSER1-AS1* [65].

*HuR-**miR-548c-3p*. Topoisomerase II alpha (*TOP2A*) is a key enzyme involved in cell proliferation, which is frequently overexpressed in cancers of different origins and influences the efficiency of the chemotherapeutic agent (e.g., doxorubicin) [66,67,68]. Recently, it has been suggested that the expression of *TOP2A* is regulated by HuR and miR-548c-3p, which competes for the binding to *TOP2A* 3′UTR [69]. Indeed, in Hela cells, HuR binds directly to mRNA (*TOP2A*) and increases its translation [69], while miR-548c-3p reduces *TOP2A* protein expression.

*HuR-miR-195.* Stromal interaction molecule 1 (STIM1) is a key regulator of store operated Ca2+ entry (SOCE), a process involved in cancer cell migration/invasion [70]. HuR promotes *STIM1* mRNA stabilization [71]. In contrast, miR-195 competes with HuR and decreases its expression, as evidenced in IEC-6 cells (normal rat intestinal crypt cells) [71].

*HuR-miR-133b.* In gastric cancer (GC) tissue and cell lines (BGC-823, MKN-45, MGC-803, SGC-7901, and AGS), HuR expression is high and promotes cancer cell proliferation and migration in vitro and in vivo [72]. This effect is mediated by the ability of HuR to decrease miR-133b expression, which exerts a tumor suppressive function in GC cells. Furthermore, HuR stabilizes *CDC5L* (cell division cycle 5-like protein) mRNA, while miR-133b inhibits its expression. Therefore, the overexpression of *CDC5L*, which promotes cell cycle progression, results from the overexpression of HuR and the loss of miR-133b in GC cells [72].

*HuR-miR-4319.* Semaphorin 4D (*SEMA4D*) is overexpressed in esophageal squamous cell carcinoma (ESCC) cells and contributes to cancer progression [73]. A competition between HuR and miR-4319 for the binding to *SEMA4D* mRNA has been demonstrated in ESCC cells (KYSE-150, TE-10 and TE-1). While HuR stabilizes *SEMA4D* mRNA and contributes to cell proliferation and migration in ESCC, miR-4319 overexpression destabilizes *SEMA4D* mRNA and prevents ESCC progression [73]. Interestingly, the expression of miR-4319 is low in ESCC cells [74], while HuR is overexpressed [75], thus indicating that *SEMA4D* overexpression in ESCC is mediated by this double hit.

*HuR-miR-300.* In gastric cancer, the expression of *UBE2C* (Ubiquitin Conjugating Enzyme E2 C), a potent tumor promoter [76], is under the regulation of HuR, which stabilizes its mRNA by preventing the binding of miR-300 [77].

*HuR-miR-494.* Nucleolin (NCL), an RBP promoting cancer cell proliferation, is regulated at the post-transcriptional level by HuR and miR-494 [78]. Indeed, in Hela cells, HuR interacts with *NCL* and promotes its translation. In contrast, miR-494 competes with HuR and represses *NCL* expression [78].

*HuR-miR-4458.* MiR-4458 is downregulated in melanoma, thus contributing to the overexpression of the pro-tumorigenic factor PBX3 (Pre-B-Cell Leukemia Transcription Factor 3) [79]. In contrast, HuR, which is overexpressed in melanoma, stabilizes *PBX3* mRNA by competing with miR-4458 [80].

*HuR-miR-4312.* In pancreatic ductal adenocarcinoma (PDAC), the overexpression of Bcl-2 associated athanogene 3 (*BAG3*) promotes IL-8 mRNA stability by promoting HuR binding to the 3′UTR [81]. The binding of HuR prevents the binding of miR-4312 and thus the downstream inhibition of IL-8 translation [81].

*HuR-miR-184.* In chronic myelogenous leukemia (CML), HuR and miR-184 compete for the binding to *MDR1* (Multidrug Resistance Protein 1) 3′UTR and thus play a major role in chemosensitivity [82]. In Adriamycin-resistant CML cells (K562), the induction of the lncRNA *FENDRR* (FOXF1 Adjacent Non-Coding Developmental Regulatory RNA) promotes miR-184 sponging, thereby favoring HuR-dependent *MDR1* mRNA stabilization [82].

*HuR-miR-675-5p.* HuR importantly contributes to cancer cell survival and aggressiveness during hypoxia by stabilizing *HIF1A* (Hypoxia Inducible Factor 1 Subunit Alpha) mRNA, as evidenced in glioma cells (U51). In contrast, miR-675-5p competes with HuR for the binding to *HIF1A* expression and promotes its downregulation [83]. These findings highlight an important mechanism involved in glioma progression.

*HuR-miR-320a.* This interplay importantly regulates the DNA damage response by controlling the expression of NONO, a RBP involved in the S phase checkpoint of the cell cycle [84]. In Hela cells, ultra-violet (UV) irradiation increases miR-320 expression in a p53-dependent manner [84] but fails to downregulate NONO expression due to the induction of HuR, which stabilizes *NONO* mRNA by competing with miR-320a. Accordingly, HuR inhibition destabilizes *NONO* mRNA, and the concomitant silencing of miR-320a restores NONO expression. These findings suggest that HuR represent an important barrier against the DNA damage response in cancer cells, which may represent an appealing therapeutic target.

*HuR-miR-124-3p.* In ovarian cancer, HuR overexpression contributes to increased stability of *NEAT1* (Nuclear Enriched Abundant Transcript 1), a lncRNA promoting cancer cell proliferation and invasion [85]. In contrast, *NEAT1* expression is suppressed by miR-124-3p [85]. However, it is unclear in this study whether HuR and miR-124-3p compete for the same binding sites. Nevertheless, it is likely that the overexpression of *NEAT1* in cancer cells is a consequence of both HuR overexpression and miR-124-3p silencing.

*HuR-miR-125b.* HuR binds to adjacent binding sites of miR-125b in the 3′UTR of P53 mRNA in MCF7 human breast carcinoma cells [86]. Upon ultraviolet C (UVC) exposure, HuR translocates from the nucleus to the cytosol, where it binds to P53 mRNA, thus preventing miR-125b-mediated translation inhibition of P53. This interplay is, therefore, important for the DNA damage response [86].

*HuR-miR-200b.* In bone marrow-derived macrophages (BMDMs) derived from myeloid-specific HuR KO mice, miR-200b expression is increased, and a computational analysis has revealed that miR-200b binding sites overlap with HuR binding sites on several transcripts [87]. Among them, HuR stabilizes *VEGFA* (Vascular Endothelial Growth Factor A) mRNA by preventing miR-200b binding. These findings demonstrate the importance of this competition in angiogenesis.

*HuR-Ptn-dt-miR-96. Ptn-dt* is an oncofetal lncRNA promoting hepatic carcinogenesis [88]. HuR, which is overexpressed in HCC, interacts and stabilizes *Ptn-dt*. Moreover, HuR directly interacts with miR-96, thereby preventing its negative effect on *Ptn-dt* expression [88].

*HuR-miR-331-3p*. In prostate cancer cells (LNCaP), HuR binds directly to the 3′UTR of *ERBB2* and stabilizes it [89], while miR-331-3p exerts an opposite function. Interestingly, HuR reduces the activity of miR-331-3p without preventing the binding of miR-331-3p on its specific site [89].

*HuR-other miRNAs.* A miRNA-binding site mapping in wild-type and HuR knockout macrophages indicate that HuR binding sites overlap with other miRNAs (i.e., miR-27) [90]. Moreover, a transcriptome-wide analysis has allowed identifying 788 HuR bound 3′UTR among the 2653 AGO2 bound 3′UTR [91]. Finally, other mechanisms have been described in chronic disorders fostering carcinogenesis, such as the competition between HuR and miR-30e for the regulation of sphingosine 1-phosphate receptor 3 (S1PR3) during liver fibrosis [92]. A potential interplay between HuR, miR-873 and miR-125a-3p has also been suggested for the regulation of cancer stemness through the regulation of CDK3 (Cyclin Dependent Kinase 3) expression in lung cancer [93]. However, the nature of this interplay was not fully depicted in this study. Finally, HuR may also affect the activity of other miRNAs through its ability to inhibit AGO2. Indeed, the circular RNA (circAGO2) generated from the *AGO2* gene can bind to HuR protein and facilitate its translocation to the cytoplasm, thereby promoting its binding to the 3′UTR of several transcripts (i.e., *EIF4EBP3*, *HNF4A*, *MAP4K1*, *NOTCH4*, *SLC2A4*, and *SLC44A4*) and reducing AGO2/miRNA-dependent regulation (miR-224-5p, miR-143-3p, miR-181a-5p, miR-503-5p, or miR-125a-3p) [94]. *CircAGO2* is overexpressed in various cancers, where it correlates with a poor prognosis and promotes tumorigenesis and aggressiveness [95]. Therefore, the *circAGO2*/HuR interplay represents a major oncogenic axis that should be considered in other malignancies.

#### 3.1.2. Other Antagonisms between HuR and miRNAs

*HuR-miR-21:* In addition to the competition between HuR and miR-21 for *PDCD4* binding, HuR directly binds and sequesters miR-21, as evidenced in MCF7 cells, thereby preventing the downregulation of miR-21 targets (i.e., *PDCD4*) [55].

*HuR-miR-7*. ALKBH5 (AlkB Homolog 5, RNA Demethylase) expression is increased in ovarian cancer tissues and cells (A2780, SKOV3 cell lines), where it contributes to cancer cell proliferation, migration, and autophagy [96]. Mechanistically, ALKBH5 increases HuR expression, which in turn downregulates miR-7 expression. MiR-7 directly targets the expression of EGFR (Epidermal Growth Factor Receptor) in SKOV3 cells [96]. Accordingly, ALKBH5 promotes EGFR expression by promoting HuR-dependent inhibition of miR-7.

*HuR-miR-16.* In colorectal cancer, cyclooxygenase 2 (COX-2), a major enzyme of inflammation, is frequently overexpressed and promotes various cancerous hallmarks [97]. This overexpression is due to the ability of HuR to directly bind and stabilize its mRNA transcript. In addition, HuR binds directly and sequesters miR-16, which directly inhibits COX-2, expression [98].

*HuR-miR-107:* In non-small cell lung cancer cells (NSCLC, A549, H1299, Calu6 and H520) ALKBH5 (AlkB homolog 5), inhibits Hippo/YAP signaling and reduces cancer cell proliferation, migration, and invasion [99]. This effect is partially promoted by an increased expression of HuR, which in turn prevents miR-107-dependent downregulation of *LATS2* (Large Tumor Suppressor Kinase 2) [99,100]. In this study, HuR binds directly to miR-107 and acts as a miRNA sponge. These findings suggest a tumor suppressive function of HuR.

*HuR-UFC1-miR-34a. UFC1* is a lincRNA overexpressed in HCC [101] and acts as a tumor promoter by favoring cancer cell proliferation, inhibiting apoptosis in vitro and in vivo. This effect is partially attributed to its ability to increase β-catenin expression by interacting with HuR. In contrast, miR-34a is a negative regulator of *UFC1* and thus prevents the induction of β-catenin in cancer cells [101].

*HuR-BBOX1-AS1- miR-361-3p.* The lncRNA *BBOX1-AS1* (BBOX1 antisense RNA 1) is overexpressed in cervical cancer, where it promotes cancer progression and metastasis formation by increasing *HOXC6* (homeobox C6) expression [102]. This effect has been associated with the capacity of *BBOX-AS1* to sponge miR-361-3p, a direct negative regulator of *HOXC6* expression [102]. In addition, HuR binds and stabilizes *BBOX-AS1*, thereby promoting the downstream inhibition of miR-361-3p. In addition, HuR binds directly and stabilizes *HOXC6*. Together, these findings indicate that the regulation of *HOXC6* is HuR and miR-361-3p-dependent, but this interplay is indirect through *BBOX1-AS1* [102].

*HuR-LINC00336-miR6852.**LINC00336* is upregulated in lung cancer and promotes cancer-related processes by acting as a competing endogenous RNA (ceRNA) on miR-6852 [103]. This miRNA is an important regulator of cystathionine-β-synthase (CBS) expression, an inhibitor of ferroptosis [103]. HuR, which is overexpressed in lung cancer, binds to *LINC00336* and stabilizes it, thus inhibiting miR-6852 and favoring CBS expression. Accordingly, miR-6852 overexpression reduces cancer cell proliferation (A549 cells) and survival by inducing ferroptosis.

*HuR-LncRNA-HGBC-miR-502-3p.* In gallbladder carcinoma, the long-non-coding RNA *HGBC* is upregulated and is associated with poor survival in patients [104]. This lncRNA importantly promotes tumorigenic processes (e.g., cancer cell proliferation, migration) [104]. HuR binds and stabilizes *HGBC* RNA and thus contributes to its overexpression. Mechanistically, *HGBC* is a miRNA sponge that directly binds to miR-503-3p, thereby preventing the downregulation of its target mRNAs, including *SET* (SET Nuclear Proto-Oncogene), an upstream regulator of AKT (Protein Kinase B) signaling [104].

*HuR-circ-CCND1-miR-646.* In laryngeal squamous carcinoma, *circ-CCND1* importantly promotes cancer-related processes by sponging miR-646, which inhibits *CCND1* (cyclin D1) expression [105]. In parallel, *circ-CCND1* interacts with HuR to stabilize *CCND1* mRNA [105]. However, these two mechanisms seem to act independently of each other, and thus the overexpression of *CCND1* is more a consequence of a coordinated effect of HuR and miR-646 rather than an interplay between them.

*HuR-CircPVT1-miR-30d/e.* In lung squamous cell carcinoma (LUSC), *circPVT1* is upregulated and correlates with a poor clinical outcome. As demonstrated in LUSC cells (A549, H520, H226, SKMES-1, and H1270), *circPVT1* acts as a ceRNA, which inhibits miR-30d, thereby promoting the expression of cyclin F, a direct target of miR-30d and miR-30e. HuR stabilizes *circPVT1* and thus indirectly contributes to the inhibition of these miRNAs [106].

*HuR-circ-FAT3-miR-136-5p.* In lung cancer cells, HuR promotes the cyclization and generation of a circular RNA, *circ-FAT3* [107], involved in various cancer-related processes. *Circ-FAT3* act as a ceRNA (competing endogenous RNA) to sponge miR-136-5p, thereby increasing the expression of its mRNA targets, including HuR [107], thus forming a regulatory loop.

*HuR-miR-199a.* HuR is an important regulator of miR-199a maturation [108] in hepatocellular carcinoma in hypoxic conditions. Hypoxia induces HuR expression and promotes its binding to the primary miR-199a (pri-miR-199a), thereby inhibiting its maturation. Due to the ability of miR-199a to regulate hexokinase-2 (*HK2*) and pyruvate kinase-M2 (*PKM2*), this interplay strongly contributes to the metabolic switch of HCC cells toward glycolysis during hypoxia [108].

*HuR-TTN-AS-1 and miR-133b*. The lncRNA *TTN-AS1* promotes cell proliferation and inhibits apoptosis in esophageal squamous cell carcinoma (KYSE410 cell line) [109]. Mechanistically, this effect is associated with its ability to sponge miR-133b, thereby preventing the downregulation of the mRNA targets of this miRNA (e.g., *FSCN1*, *SNAIL*, N-cadherin, Vimentin). Moreover, *TTN-AS1* recruits HuR, which, in turn, stabilizes *FSCN1* (Fascin Actin-Bundling Protein 1) and promotes epithelial–mesenchymal transition. Finally, HuR directly reduces miR-133b expression through a poorly characterized mechanism [109].

*HuR-**circ_0036412-miR-579-3p*. In hepatocellular carcinoma, the circular RNA *circ_0036412* increases *GLI2* (*GLI Family Zinc Finger 2*) expression, which promotes the Hedgehog pathway and cancer cell proliferation [110]. Mechanistically, *circ_0036412* sponges miR-579-3p to prevent *GLI2* downregulation and recruits HuR to stabilize *GLI2* transcript, as evidenced in HCC cells (Huh-7 and Hep3B cell lines) [110].

*HuR-HOTAIR and miR-7.* In Neck Squamous Cell Carcinoma, the lncRNA Homeobox (*HOX*) transcript antisense RNA (*HOTAIR*) is overexpressed, and this effect is partially mediated by HuR, which directly binds and stabilizes *HOTAIR* [111]. *HOTAIR* acts as a miRNA sponge on miR-7 to promote cancer cell (CC25 and FaDu cells) proliferation, migration, and invasion.

*HuR-H19 and miR-675.* HuR interacts directly with the lncRNA *H19* in the placenta and prevents its processing for the biogenesis of miR-675 [112]. This inhibitory effect occurs during the processing by drosha [112]. However, it is unclear whether this link exists in human cancers. Moreover, depending on the cancer type, miR-675 expression is reduced or overexpressed and displays both oncogenic (e.g., Non-Small Cell lung cancer, [113] and tumor-suppressive properties (e.g., prostate cancer, [114]). In contrast, H19 is upregulated and promotes cancer progression. Therefore, the induction of HuR, frequently observed in human cancer, may inhibit miR-675 expression in cancer cells. However, this link remains to be determined.

#### 3.1.3. Cooperation between HuR and miRNAs

*HuR-let-7.* HuR can potentiate the effect of some miRNAs, as exemplified by let-7, which binds to the 3′UTR of c-Myc more efficiently when HuR binds to a proximal ARE site [115,116]. This interplay promotes a translational repression of c-Myc due to the ability of HuR to promote the interaction between AGO2/let-7b/c with c-Myc mRNA. Given the oncogenic potential of c-Myc [117], this study questions the dogmatic view of HuR as a strict oncogenic/tumor-promoting factor. Interestingly, both the expression of HuR and c-Myc are enhanced in most cancers, while the expression of let-7b and c is frequently decreased [118,119,120,121]. These findings suggest a decoupling between HuR and Let-7 in cancer cells, thereby preventing the ability of HuR to inhibit the translation of oncogenic transcripts (e.g., c-Myc). Re-expressing let-7 expression may, therefore, represent a potent therapeutic approach by restoring the capacity of HuR to inhibit the translation of several transcripts. The cooperation between HuR and let-7b/RISC, favoring tumor progression, has also been described in HeLa cells. HuR recruits let-7b and both binds to and downregulates *lincRNA p21*. *LincRNA p21* directly binds and prevents the translation of β-catenin and *JUNB* [122]. Therefore, the HuR-let-7b axis favors β-catenin and *JUNB* protein expression in cancer cells.

*HuR-miR-9.* In Hodgkin lymphoma, miR-9 directly represses several cytokines, including IL-6 (interleukin-6), TNFα (tumor necrosis factor-alpha), and CCL-5 (C-C Motif Chemokine Ligand 5). Interestingly, the regulation of IL-5 (interleukin-5) and TNFα is HuR-dependent. Indeed, the knockdown of HuR in HEK293T cells increases the expression of these cytokines, and this effect is prevented by the inhibition of miR-9. Although the underlying mechanism has not been fully depicted, these data suggest that HuR and miR-9 cooperate for the regulation of these cytokines [123].

*HuR-miR-19a and b.* MiRNAs bind to their targets through a specific base pairing between their seed sequence and the target mRNAs [17,18,19]. However, increasing evidence indicates a non-canonical mechanism of miRNAs binding, which is mediated by HuR. More precisely, in breast cancer cells, *ABCB1* (P-glycoprotein), a multidrug resistance protein, is regulated by miR-19b despite the absence of miR-19b binding sites within the 3′UTR of *ABCB1* mRNA [124]. The binding of miR-19 is mediated by HuR [124], thus suggesting that this cooperation importantly controls the chemosensitivity of cancer cells. Interestingly, HuR is also important for miR-19-dependent regulation of Ras homolog B (RhoB) in keratinocytes upon UV exposure [125]. Finally, in papillary thyroid cancer cells (BCAP) and HEK-293T cells, HuR overexpression promotes miR-19a synthesis, while its silencing reduces miR-19a levels [126]. This effect is mediated by the binding of HuR to pre-miR-19a, thereby increasing its stability and favoring its maturation. These findings further indicate that HuR is not only an mRNA stabilizing factor. Interestingly, this study suggests that the HuR-miR-19a/b axis promotes cancer cell proliferation. However, miR-19a targets Ras-related RAP-IB (*RAP1B*), a GTPase member of the Ras-associated protein family (RAS), which is overexpressed in BCAP and promotes cancer cell proliferation, migration, and invasion [127]. Further investigations are, therefore, required to fully characterize the importance of the HuR/miR-19a axis in carcinogenesis.

*HuR-miR-1246.* In gastric cancer cells (AGS cells), HuR binds to an ARE present on miR-1246 and promotes its secretion in the exosome [128]. Although the underlying mechanism is still unclear, exsosomal miR-1246 is frequently observed in the serum of patients with GC and may contribute to various cancerous processes [129,130,131].

*HuR-miR-200a.* MiR-200a and HuR cooperate to stabilize c-Jun mRNA, as evidenced in HEK293T cells [132]. This study challenges again the dogmatic view of miRNAs as strict gene expression repressors and indicates that this mechanism can contribute to the overexpression of important oncogenes.

*HuR-**miR-200c.* In ovarian cancer, the role of miR200c depends on the localization of HuR. Indeed, when HuR is in the nucleus, miR200c acts as a tumor suppressor by inhibiting the expression of *TUBB3* (class III β-tubulin) mRNA. In contrast, when HuR localizes in the cytoplasm, miR200c binds to HuR and acts in concert to stabilize *TUBB3* mRNA, thus promoting carcinogenesis [133]. Together, these data further indicate that HuR exerts a dual function depending on the context, which influences HuR localization.

*HuR-miR-494.* In muscle invasive bladder cancer (MIBC) cells (T24T), the anti-cancerous compound ChlA-F, a novel C8 fluoride derivative of cheliensisin A, increases *JUNB* mRNA stability in a HuR-dependent manner [134]. Then, Jun-B promotes miR-494 expression, which, in turn, directly binds and decreases c-Myc mRNA [134].

*HuR-circular RNAs.* Increasing evidence indicates an interplay between HuR and lncRNAs, which is not limited to their ability to sponge miRNAs. The *circZNF609* RNA interacts with HuR to stabilize *CKAP5* (Cytoskeleton Associated Protein 5) mRNA [135], a cytoskeleton/mitosis-associated factor, in RD cells (embryonal rhabdomyosarcoma cell line). Similarly, HuR and *AGAP2-AS* interact and stabilize *CPT1A* (Carnitine Palmitoyltransferase 1A) in mesenchymal stem cells, thereby promoting trastuzumab resistance in breast cancer [136]. *CCAT2* (Colon Cancer Associated Transcript 2), a lncRNA, interacts with HuR in the nucleus to promote HCC progression, potentially by regulating the expression of autophagy-related genes [137]. Finally, HuR interacts with *circCCNB1* and miR-516b-5p and cooperates to stabilize cyclin-D1 (*CCND1*) mRNA in glioma, thereby favoring cancer progression [138].

*HuR-RPSAP52-miR-15a, miR-15b, and miR-16**.* This interplay between HuR, the lncRNA *RPSAP52* (Ribosomal Protein SA Pseudogene 52) and miRNA-dependent regulation has been associated with the regulation of the cell cycle inhibitor, p21, in colorectal cancer cells (HT-29) [139]. In this study, *RPSAP52* directly binds to HuR protein, thereby preventing its ability to stabilize *CDKN1A* (Cyclin Dependent Kinase Inhibitor 1A) transcript. In addition, *RPSAP52* shifts the localization of miR-15a, miR-15b and miR-16 from *HMGA* (High Mobility Group AT-Hook 1) to *CDKN1A* mRNA. One potential explanation discussed in this study is the change in mRNA conformation, thus allowing better access of miRNA to *CDKNA1*.

*HuR-other miRNAs.* Although HuR has mainly been associated with a stabilizing function, this dogmatic view is further challenged by other studies showing cooperation between HuR and miRNAs in other diseases. For instance, HuR and miR-26 act synergistically to regulate the expression of the negative regulator of G-protein signaling 4 (*Rgs4*) in neurons [140]. Whether this cooperation exists in human cancers remains to be determined, but these findings suggest that this destabilizing function of HuR is underestimated in cancers and may be tissue dependent.

### 3.2. TTP Family Members (TTP, TIS11, GOS24, NUP475) 

TTP is one of the most studied AUBPs, which belongs to the family of Cys-Cys-Cys-His zinc finger proteins [141,142,143]. This family contains three members, TTP (tristetraprolin, *ZFP36*), BRF1 (Butyrate response factor 1, *ZFP36L1*) and BRF2 (Butyrate response factor 2, *ZFP36L2*) [143]. TTP is rapidly induced by various stimuli, including TGF-β (transforming growth factor-beta) [144,145], LPS (lipopolysaccharide) [146], TNFα, or insulin [147,148] and is, therefore, considered as an immediate–early response gene [9]. TTP is one of the best-characterized AUBPs involved in ARE-mediated mRNA decay through a process involving the nucleation of small cytoplasmic granules called (P-bodies), where the target mRNAs are bound by various enzymes promoting deadenylation, decapping and degradation [9,149]. The importance of TTP in physiological/pathological processes is illustrated by *Zfp36* knockout mice, which develop a severe inflammatory syndrome and growth retardation due to the overexpression of pro-inflammatory factors (e.g., TNF-α) [9,150,151]. TTP is mostly considered a tumor suppressor whose expression is lost in tumors. This effect is due to its ability to control a whole network of oncogenic and pro-inflammatory transcripts [9]. However, this concept has recently been challenged in HCC, where TTP plays a potent pro-tumorigenic function in vivo, as evidenced in liver specific TTP KO mice treated with a hepatic carcinogen (i.e., diethylnitrosamine) [152]. The interplay between TTP and miRNA-dependent regulation is poorly known and even less understood in the context of cancer. TTP can cooperate with some miRNAs, such as miR-16, as evidenced in drosophila S2 cells [153]. MiR-16 sequence (UAAAUAUU) is complementary to the canonical ARE sequence. This cooperation requires the binding of TTP to AGO/EIF2C family members to assist miR-16 binding to ARE on its target [153] mRNAs. Moreover, TTP can also promote the expression of some miRNAs, as evidenced by let-7, whose expression is increased by ectopic overexpression of TTP in various cancer cells (PA1, HCT-116) [154]. This effect leads to the downregulation of let-7 target genes, such as *CDC34* (Cell Division Cycle 34), and reduces cancer cell proliferation [154]. Interestingly, this effect seems to be P53-dependent, as DNA-damaging agents such as doxorubicin trigger P53 activation and induce both TTP and let-7 expression [155]. These data also indicate that the expression of TTP is tightly dependent on the mutational landscape in cancer cells. However, it is currently unclear whether the expression of TTP is similarly affected by other mutations. Whether a specific mutation is associated with a specific post-transcriptional signature remains to be determined. Both TTP and let-7 are frequently downregulated in human cancers. Restoring TTP expression may, therefore, represent a potential approach to restoring let-7 expression in cancer cells. Several approaches aiming at restoring TTP expression in cancer cells have been proposed, such as DNA demethylating agents in hepatocellular carcinoma or histone deacetylase inhibitors (HDAC) in colorectal cancer and HCC [152,156]. In ovarian cancer cells (CaOV-3, SKOV-3, HEYA-8 cell lines), p70S6K (70 S6 kinase) phosphorylates and inhibits the interaction of TTP with Dicer, thereby reducing the expression of miR-145 [157]. Other studies have also demonstrated that TTP can reduce the expression of miRNAs, such as miR-155 in cystic fibrosis [158]. This effect is mediated by the induction of miR-1, which reduces miR-155 biogenesis in lung epithelial cells. Collectively, these findings indicate that TTP can indirectly control the expression of a myriad of transcripts by controlling the expression of miRNAs. This interplay is currently poorly known for the other TTP family members, BRF1 and BRF2. In lung cancer, the lncRNA *MNX1-AS1* promotes cancer progression by interacting with miR-527, thereby preventing miR-527-induced BRF2 downregulation. In this study, BRF2 importantly promotes cancer cell proliferation, migration, and invasion [159].

## 4. MiRNAs-Regulating HuR and TTP Family Members

Several miRNAs targeting AUBPs have been documented in different cancers. Targeting these miRNAs may, therefore, represent a potential therapeutic approach aiming at restoring the “beneficial” AUBPs. In this section, we discuss the miRNAs regulating the expression of the most studied AUBPs, TTP family members and HuR (Table 1 and Figure 3).

### 4.1. HuR

Several miRNAs regulating HuR expression have been identified in human cancers. Although HuR expression is increased in most neoplasms [160], this knowledge is limited to certain types of cancers and has essentially been demonstrated in vitro using cancer cell lines.

*Gastric Cancer (GC):* HuR expression is increased in GC as compared to normal tissue and contributes to tumor growth and metastasis [161]. This induction of HuR can be attributed to the loss of expression/activity of several miRNAs, including miR-145 [161], miR-519 [162] or miR-582-3p [163]. Interestingly, the activity of miR-582-3p is inhibited by a circular RNA, *circSHKBP1*, which is upregulated in GC. These data illustrate well the complex regulatory network regulating HuR expression, with various miRNAs and ncRNA acting in concert to increase HuR expression. Finally, not only the expression but also the cytoplasmic localization of HuR is under miRNAs influence in GC. Indeed, GAS5 (Growth Arrest Specific 5) directly binds to HuR and promotes its cytoplasmic translocation, thereby stabilizing the *FAM83B* (Family With Sequence Similarity 83 Member B) transcript and promoting cancer cell proliferation, migration, and invasion. MiR-140-3p, which is downregulated in GC patients, directly binds and inhibits the expression of *GAS5*, and thus, promotes HuR nuclear localization and the downregulation of *FAM83B* expression [164].

*Hepatocellular Carcinoma (HCC).* MiR-16 is a potent regulator of cyclooxygenase-2 expression in human hepatoma/HCC cells (Hep3B, WRL68) [165]. This effect is mediated by two different mechanisms with (*i*) a direct regulation of COX-2 mRNA by miR-16 and (*ii*) the ability of miR-16 to downregulate HuR expression, which stabilizes COX-2 mRNA. MiR-16 exerts tumor suppressive properties in vitro and in vivo (xenograft model), and its expression inversely correlates with COX-2 level in HCC tissue [165].

*Ovarian cancer.* The upregulation of HuR in ovarian cancer is also associated with the downregulation of miRNAs, such as miR-139-3p, whose loss promotes tumor growth and metastasis of ovarian cancer cells [166]. Moreover, similarly to GC, miR-519 is also downregulated in ovary, lung, and kidney cancer, and miR-519 overexpression reduces tumor growth by reducing HuR expression [167,168].

*Papillary thyroid carcinoma.* In this cancer, only miR-31 downregulation has been associated with HuR overexpression [169]. In thyroid carcinoma, HuR is an important tumor promoting factor due to its ability to stabilize oncogenic transcripts [170], and thus, a better understanding of the post-transcriptional regulation of HuR is required to envisage novel therapeutic options.

*Non-Small Cell Lung Carcinoma (NSCLC).* NSCLC is a leading cause of cancer mortality [171]. As described for ovarian cancer, miR-139-3P downregulation contributes to HuR overexpression in lung cancer cells [172]. In addition, other miRNAs, including miR-146A-5p [173], miR-31 [174], miR-136-5p [107], or miR-519 [167], have been recognized as direct regulators of HuR, which are downregulated/inhibited in tumors.

*Colorectal cancer.* HuR is upregulated in CRC and this effect has been associated with the downregulation of various miRNAs, including miR-22 [175], miR-324-5P [176], miR-519 [168,177], miR-155-5p [178] or miR-34b-5p [60]. In addition, HuR overexpression has been associated with *TNFRSF10A-AS1*; a lncRNA overexpressed in CRC, which binds and inhibits miR-3121-3p, a direct regulator of HuR [179]. Together, these studies illustrate that HuR upregulation depends on a concerted effect of all these miRNAs, whose activity depends on lncRNAs.

*Cervical cancer.* Similarl to CRC, miR-324-5P and miR-519 are potent regulators of HuR expression in cervical cancer, which are downregulated in tumors and associated with radiotherapy resistance to cancer cells [168,180,181].

*Rhabdomyosarcoma (RMS).* In RMS cells (Rh30 cells), miR-29 interacts directly with HuR and acts as a decoy to prevent HuR binding to *A20* mRNA (TNF Alpha Induced Protein 3), an upstream repressor of NFkB (Nuclear factor kappa B) [182]. In these cells, HuR overexpression leads to *A20* downregulation. In RMS and osteosarcoma cells, as well as in tumor samples from patients (e.g., liposarcoma, RMS, osteosarcoma), the expression of *A20* and miR-29 is low, while HuR is strongly upregulated. The decoy activity of miR-29 may reside in the precursor sequence of the miR, which contains putative binding sites for HuR. Together, these results provide further evidence that HuR can also function as an RNA-destabilizing factor.

*Glioblastoma multiforme:* miR-3127-5P is a direct regulator of HuR in glioma [183]. Interestingly, this miRNA is sponged by LncRNA gastric cancer-associated transcript 3 (*GACAT3*), which is upregulated in glioma tissues from patients and thus promotes HuR overexpression [183]. In glioma stem cells (GSCs), [184], miR-146b-5p is also a direct regulator of HuR, and its downregulation in glioma promotes HuR expression, which, in turn, negatively regulates the expression of *LincRNA-p21*, a long intergenic non-coding RNA involved in the inhibition of β-catenin expression and nuclear translocation in GSC [184].

*Breast cancer:* Knockdown of miR-16 in MDA-MB-231 human breast carcinoma increases HuR expression [185]. MiR-16 directly interacts with the 3′UTR of HuR [185]. Other miRNAs regulating HuR have been identified in breast carcinoma, including miR-125A [186] but also miR-519 [187]. Interestingly, miR-29a can also regulate HuR indirectly through its ability to downregulate TTP. Indeed, HuR mRNA contains ARE in its 3′UTR, which are recognized by TTP. Therefore, miR-29a inhibition restores TTP expression, which, in turn, downregulates HuR [188].

*Prostate cancer.* MiR-133b is a potent regulator of HuR, which is downregulated in prostate cancer cells (PC3, DU-145 cell lines) [189]. Accordingly, overexpression of miR-133b decreases HuR expression through a direct binding to HuR 3′UTR. This negative impact on HuR expression induced docetaxel cytotoxicity in prostate cancer cells [190] by reducing the expression of *ABCG2* (ATP Binding Cassette Subfamily G Member 2), a transporter involved in chemoresistance. HuR expression is also negatively regulated by miR-34a in paclitaxel-resistant PC3 cells (PC3PR). The downregulation of HuR by miR-34a leads to a decrease in *SIRT1* (Silent mating type information regulation 2 homolog 1) and Bcl-2, thereby sensitizing cancer cells to paclitaxel [191].

*Nasopharyngeal Carcinoma (NPC).* MiR-514-5p is a direct regulator of HuR expression in NPC cells (CNE1, CNE2, C666–1 and HNE1 cell lines) [192]. Interestingly, the activity of this miRNA is negatively regulated by the lncRNA *SNHG7*, which is upregulated in NPC cells [192].

*Oropharyngeal Squamous Cell Carcinoma (OPSCC).* MiR-133a-3p is downregulated in OPSCC smokers, Human Papilloma Virus (HPV) positive cells and in E6/E7 overexpressing HPV negative cells treated with cigarette smoke extract [193]. MiR-133a-3p knockdown leads to an upregulation of HuR [193].

*Laryngeal squamous cell carcinoma.* MiR-519a is a potent tumor suppressor in laryngeal squamous cells [194]. This effect is partially mediated by its ability to directly inhibit HuR expression, as evidenced in laryngeal squamous cell carcinoma human epithelial type 2 cells [194]. Interestingly, HuR is also directly regulated by miR-519b-3p in larynx squamous Hep-2 cells, and the expression of this miRNA is reduced in tumors from patients [195].

*Esophageal Squamous Cell Carcinoma (ESCC)*. In ESCC cells (KYSE410), miR-133b inhibits HuR expression and prevents *FSCN1* (Fascin Actin-Bundling Protein 1) expression and epithelia-mesenchymal transition [109]. In patients, miR-133b loss correlates with a poor prognosis and contributes to HuR overexpression, which promotes cancer cell proliferation.

*Hematologic malignancies.* MiR-519 regulates HuR expression in HL60 cells (acute myeloid leukemia, AML) [196]. Overexpression of miR-519 in AML cells reduces cancer cell proliferation by reducing HuR expression, thus indicating that HuR is not only a pro-tumorigenic factor. Other miRNAs have been involved in HuR regulation in AML, including miR-199a-3p [197] and miR-25. However, this latest is an indirect regulator involving the regulation of the NOX4 (NADPH Oxidase 4)/JNK (JUN N-Terminal Kinase) axis involved in HuR phosphorylation and cytoplasmic translocation [198]. In multiple myeloma cells, miR-16 downregulation correlates with the overexpression of HuR [199]. These results have been demonstrated by using a statistical approach (LIMMA method). In T-Large Granular Lymphocyte Leukemia (T-LGLL), miR-146b is a regulator of HuR expression, which is downregulated in a STAT3 (Signal Transducer and Activator of Transcription 3)-dependent manner [200]. Finally, in a model of Hodgkin lymphoma (L428, L540 and KM-H2 cell lines), miR-9 directly targets HuR and inhibits its expression [123]. Accordingly, the use of miR-9 antagomiR promotes HuR upregulation [123]. In B lymphoma cells (RAMOS, Daudi), miR-17-19b binds and reduces MYC expression [201]. This effect is indirectly mediated by the downregulation of checkpoint 2 expression (*CHK2*), an upstream regulator of HuR phosphorylation. The decreased HuR phosphorylation promotes HuR binding to *MYC*, which reduces its translation [201].

*Other potential miRNAs regulating HuR?* The number of miRNAs regulating HuR expression is probably underestimated given that several miRNAs have been predicted in several databases (e.g., miRwalk: http://mirwalk.umm.uni-heidelberg.de/: accessed on 13 June 2022) but have not been validated yet (Figure 3B,D). Therefore, it is likely that the overexpression of HuR in cancers may also result from a concerted effect of all these miRNAs.

### 4.2. TTP Family Members

#### 4.2.1. Tristetraprolin (*ZFP36*)

TTP is mostly associated with tumor suppressive properties in human cancers, and accordingly, its expression is frequently reduced in tumors [143]. Although various mechanisms have been described (e.g., promoter methylation, constitutive protein degradation, HDAC-dependent silencing of transcription factors) [156,202], several miRNAs have been involved in its loss. Among them, miR-29a is a direct regulator of TTP, which is upregulated in breast cancer [188,203]. MiR-29a is also upregulated in pancreatic cancer [204] and promotes cancer cell proliferation and migration by directly targeting TTP, thereby increasing the expression of epithelial-mesenchymal transition (EMT) and inflammatory markers. Whether this link between miR-29a and TTP exists in other cancers remains unclear, but the expression of this miRNA is frequently increased in human cancers [205], while TTP expression is lost. MiR-182 is another well-known miRNA able to downregulate the expression of TTP in cancers, which is frequently overexpressed in tumors [206]. Interestingly, this miR-182/ZFP36 axis is also regulated by the circular RNA_00054 (*circRNA_00054*), which acts as a miRNA sponge against miR-182 [206]. Accordingly, overexpression of *circRNA_00054* in breast cancer cells reduces miR-182 bioavailability, restores TTP expression and promotes cancer cell growth, invasion, and migration in vitro and in vivo. MiR-128a, which is upregulated in Acute Myeloid Leukemia (AML), is also a potential regulator of TTP expression, as demonstrated in AML cells (OCI-AML3 and APL/AML cells), where its knockdown strongly triggers TTP expression [207]. However, it is unclear in this study whether this effect is direct or not. In hepatocellular carcinoma, miR-9-5p is a direct inhibitor of TTP expression [208]. Interestingly, the transfer of exosomal SENP3-EIF4A1 inhibits tumor growth in vivo by sequestering miR-9-5p, thereby increasing TTP expression [208]. Finally, miR-200c was also identified as a direct regulator of TTP expression in mouse breast cancer cells (4TO7) [209].

Together, these findings strongly suggest that TTP loss is promoted by the concerted impact of several miRNAs on TTP mRNA. Like for HuR, the number of miRNAs regulating TTP expression is probably underestimated given that many other miRNAs are predicted to target *ZFP36* mRNA but have not been validated yet (Figure 3B, D).

#### 4.2.2. Butyrate Response Factor 1 (*ZFP36L1*)

In contrast to HuR and TTP, very few miRNAs involved in *ZFP36L1* regulation have been identified. Among them, miR-93-3P is strongly increased during wound healing in mice and contributes to keratinocyte proliferation by targeting *ZFP36L1* mRNA. [210]. In glioblastoma multiforme (GBM) cells, overexpression of miR-129-5p reduces *ZFP36L1* expression and impairs cancer cell proliferation, migration, and invasion [211]. High expression of *ZFP36L1* in GBM patients correlates with a poor clinical outcome and correlates with the downregulation of miR-129-5p. Finally, miR-181b was suggested as a potential regulator of *ZFP36L1* in Chronic Lymphocytic Leukemia (CLL). However, no validation experiments were provided in this study, and thus, the link between this miRNA and BRF1 remains to be firmly demonstrated [212].

#### 4.2.3. Butyrate Response Factor 2 (*ZFP36L2*)

The post-transcriptional regulation of BRF2 is almost unknown, but emerging evidence indicates that some miRNAs are also involved, such as miR-375 in pancreatic ductal adenocarcinoma [213]. In this study, the authors demonstrated that the expression of miR-375 is strongly reduced in PDAC from patients. Furthermore, overexpression of miR-375 in PDAC cells (i.e., PANC-1 and SW1990) strongly refrains cancer cell proliferation, migration, and invasion. This effect is partially attributed to the decrease in *ZFP36L2*, which is a direct target of miR-375. Finally, in patients, high expression of *ZFP36L2* is associated with a poor clinical outcome. In NSCLC, miR-373, which is a direct regulator of BRF2 [214], is downregulated. The loss of miR-373 and the overexpression of BRF2 correlates with a poor prognosis in patients. Accordingly, miR-373 overexpression or BRF2 knockdown in cancer cells (A549 cells) impairs proliferation, migration, and invasion [214]. Similar findings were obtained in lung adenocarcinoma with let-7b-3p, which is also downregulated and is a direct regulator of BRF2 expression [215].

**Table 1 cancers-14-03516-t001:** MiRNAs regulating HuR and TTP family members and involved in cancers. n.a: not available.

miRNA	Targets (AUBPs)	Models Used	Expression/Activity in Patients	Reference
**Gastric cancer**				
**miR-145**	*ELAVL1*	SK-OV-3, A2780 and OVCAR-3	Down	[161]
**miR-519**		n.a	Down	[162]
**miR-582-3p**		HGC27, BGC823 cells	Down	[163]
**Ovarian cancer**				
**miR-139-3p**	*ELAVL1*	SK-OV-3, A2780 and OVCAR-3	Down	[166]
**miR-519**		Hela, A21780 and HOSE-B cells	Down	[167]
**Papillary thyroid carcinoma**				
**miR-31**	*ELAVL1*	Ovarian carcinoma cells	Down	[169]
**Non-small cell lung carcinoma**				
**miR-139-3p**	*ELAVL1*	Normal bronchial epithelial cells (BEAS-2B) and NSCLC cells (H1299, H1975, HCC827, H1650 A549)	Down	[172]
**miR-146a-5p**		A549 cells	Down	[173]
**miR-31**		n.a	Down	[174]
**miR-519**		Hela, A21780 and HOSE-B cells	Down	[167]
**miR-373**	*ZFP36L2*	A549 cells	Down	[214]
		**Lung adenocarcinoma**		
**Let-7b-3p**	*ZFP36L2*	H1299, A549 cells	Down	[215]
		**Hepatocellular carcinoma**		
**miR-16**	*ELAVL1*	Hep3B, WRL68	Down	[165]
**miR-9-5p**	*ZFP36*	HuH7, Hep3B	Down	[208]
**Colorectal cancer**				
**miR-22**	*ELAVL1*	NCM460, SW480, HT29, HCT15, HCT116, SW620, Caco2, LOVO	Down	[175]
**miR-324-5p**		SW620, SW480, NCM-460 cells	Down	[176]
**miR-519**		S1, S1M1 80 (mitoxantrone-resistant), Caco-2, HT-29, SW620	Down	[168,177]
**miR-155-5p**		HT-29	Up	[178,216]
**miR-34b-5p**		NCM460, SW620, HT-29, HCT116, LoVo, RKO	Down	[60]
**miR-3121-3p**		DLD-1, HCT116, HT29, SW480	Down	[179].
**Prostate cancer**				
**miR-133b**	*ELAVL1*	PC3, DU-145 cell lines	Down	[189,217,218]
**miR-34a**		PC3	Down	[191,219]
**Cervical cancer**				
**miR-324-5p**	*ELAVL1*	33A, ME-180, Hela and Caski	Down	[168,180]
**miR-519**		Hela cells	Down	[168,180]
**Pancreatic cancer**				
**miR-29a**	*ZFP36*	Panc-1, HPDE6c7, BXPC-3	Up	[204]
**miR-375**	*ZFP36L2*	Pancreatic ductal adenocarcinoma PANC-1 and SW1990	Down	[213]
**Glioblastoma**				
**miR-3127-5p**	*ELAVL1*		Down	[183]
**miR-146b-5p**		Glioma stem cells	Down	[184]
**miR-129-5p**	*ZFP36L1*	LN229, A172, U87, T98G, U251, H4, LN118 and normal astrocytes	Down	[211]
**Breast cancer**				
**miR-16**	*ELAVL1*	MDA-MB-231	Down	[185]
**miR-125a**		MCF-7	Down	[186,220]
**miR-29a**	*ELAVL1* *(indirectly)*	MDA-MB-231, MCF-7, MCF12A, MCF10A cells	Up	[188]
**miR-519**	*ELAVL1*	MCF-7 cells	Down	[187]
**miR-29a**	*ZFP36*	MDA-MB-231 and MCF-7, MCF12A and MCF10A (normal-like breast cell line), and HEK293 kidney cells	Up	[188,203]
**miR-182**	*ZFP36*	MDA-MB-231, SUM-159, MCF-7, SK-BR-3, MDA-MB-157	Up	[206]
**miR-200c**	*ZFP36*	4TO7 cells	Up	[209,221]
**Nasopharyngeal carcinoma**				
**miR-514-5p**	*ELAVL1*	CNE1, CNE2, C666–1 and HNE1 cells	Down	[192]
**Oropharyngeal squamous cell carcinoma**				
**miR-133a-3p**	*ELAVL1*	UMSCC47 and UMSCC11A cells	Down	[193]
**Laryngeal squamous cell carcinoma**				
miR-519a	*ELAVL1*	laryngeal squamous cell carcinoma human epithelial type 2 cells	Down	[194]
**Hematologic malignancies**				
**miR-519**	*ELAVL1*	HL-60 cells (AML)	Up	[196]
**miR-25**	*ELAVL1 (indirectly)*	U937 (AML)	Down	[198,222]
miR-199a-3p	*ELAVL1*	Bone marrow cells	n.a.	[197]
miR-25	*ELAVL1* *(indirect)*	Bone marrow cells	Down	[197,222]
**miR-146b**	*ELAVL1*	T-large granular lymphocyte leukemia (T-LGLL),	Down	[200]
**miR-128a**	*ZFP36*	AML cells (OCI-AML3 and APL/AML)	Up	[207]
**miR-181b**	*ZFP36L1 (not validated)*	CLL cells	Down	[212]
**miR-9**	*ELAVL1*	Hodgkin lymphoma L428 cells	Up	[123,223]

## 5. Conclusions

In this review, we discussed several mechanisms associated with HuR and TTP family members, which indicate that these proteins form a complex regulatory network with non-coding RNAs, that can exert both oncogenic and tumor-suppressive functions. This level of complexity between these post-transcriptional regulators is poorly considered in human diseases but also in physiological processes (e.g., development), and thus, our current interpretation of their role in cancers is mostly incomplete and probably misleading. Moreover, emerging evidence indicates that these crosstalks also occur in preneoplastic stages, including chronic inflammatory and metabolic disorders, and thus may contribute to the onset of cancers [224,225]. These findings challenge the dogmatic function of some AUBP (e.g., HuR) and miRNAs (e.g., miR-21) in cancers and clearly indicate that miRNA activity is not reflected by their deregulated expression only but also their bioavailability. Taken together, these findings suggest cautions regarding the therapeutic approaches aimed at targeting miRNAs or AUBPs (e.g., HuR-specific inhibitors suggested for the treatment of cancers) [51,226]. Therefore, a better understanding of this regulatory network and of the impact of these therapeutic approaches on miRNA-dependent regulation of gene expression is needed prior to any clinical application. In this regard, one of the major issues is the current models available to study human diseases, which do not fully recapitulate the complexity of these interactions occurring in patients. Therefore, intense efforts are required to integrate these networks in more physiological models (in vivo) rather than assessing the role of individual regulators in vitro. In this review, we discussed the crosstalks between the AUBPs, HuR, TTP family members, and miRNAs, but it is likely that a higher level of complexity is occurring with other RBPs, lncRNA, ceRNA, and RNA editing. Although modeling this entire network is currently challenging, new bioinformatics tools (database repository, deep learning, artificial intelligence) [227] will represent an asset in the future to achieve this goal and to identify new players of carcinogenesis, which may represent novel therapeutic targets. Finally, identifying the miRNAs/lncRNA regulating AUBPs and thus important oncogenes and tumor suppressors is also of major importance for designing novel therapeutic approaches. The current findings indicate that this regulation depends on several miRNAs regulating HuR and TTP family members in concert. Targeting the most consistent mechanisms (e.g., miR-519 for HuR) may also represent an appealing approach for different cancers.

## Figures and Tables

**Figure 1 cancers-14-03516-f001:**
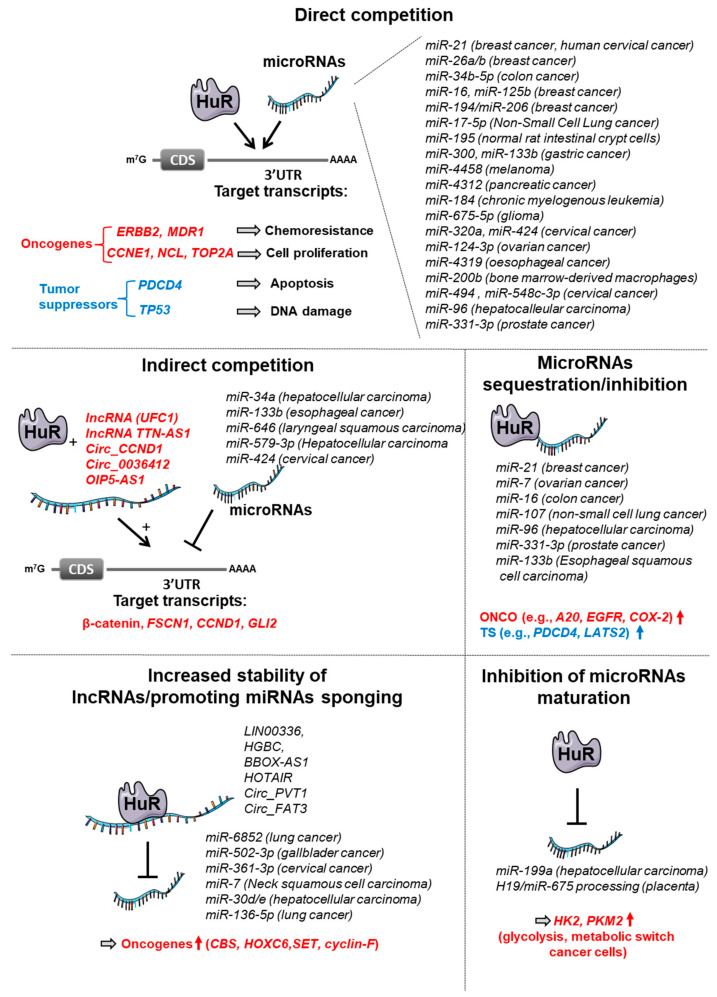
Antagonisms between HuR and microRNA-dependent regulation. HuR can interfere with miRNA-dependent regulation by different mechanisms: (*i*) HuR and miRNAs can compete for the binding to mRNAs due to overlapping binding sites; (*ii*) HuR, in concert with lncRNAs, can indirectly compete with miRNAs; (*iii*) HuR can directly sequester or inhibit miRNA expression; (*iv*) HuR can inhibit miRNA maturation (i.e., miR-199a); (*v*) HuR can promote lncRNA-dependent miRNA sponging. ONCO: Oncogenes; TS: Tumor Suppressor.

**Figure 2 cancers-14-03516-f002:**
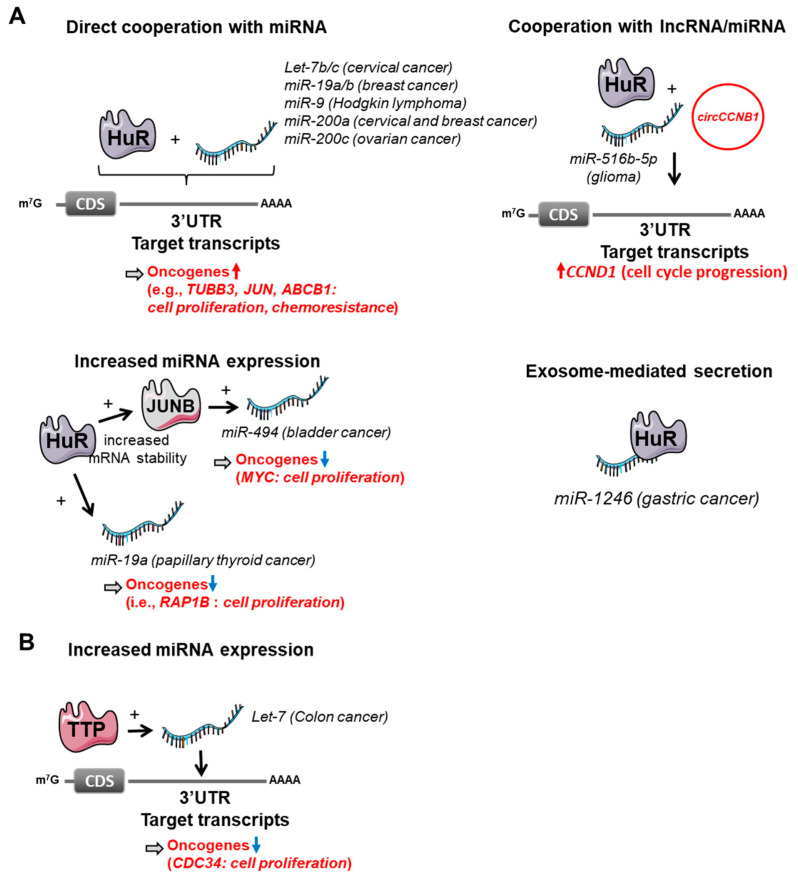
Cooperation between HuR, TTP and microRNA-dependent regulation. (**A**) HuR can cooperate with miRNAs by directly interacting with miRNAs and favoring their binding to their target mRNAs. This effect can also require the binding of circRNA (e.g., *circ_CCNB1*). HuR promotes exosome-dependent miR-1246 secretion. HuR can stabilize *JUNB* mRNAs, thereby favoring its overexpression, which in turn promotes miR-494 expression. (**B**) Finally, TTP can promote let-7 expression, which, in turn, decreases its mRNA targets (e.g., *CDC34*).

**Figure 3 cancers-14-03516-f003:**
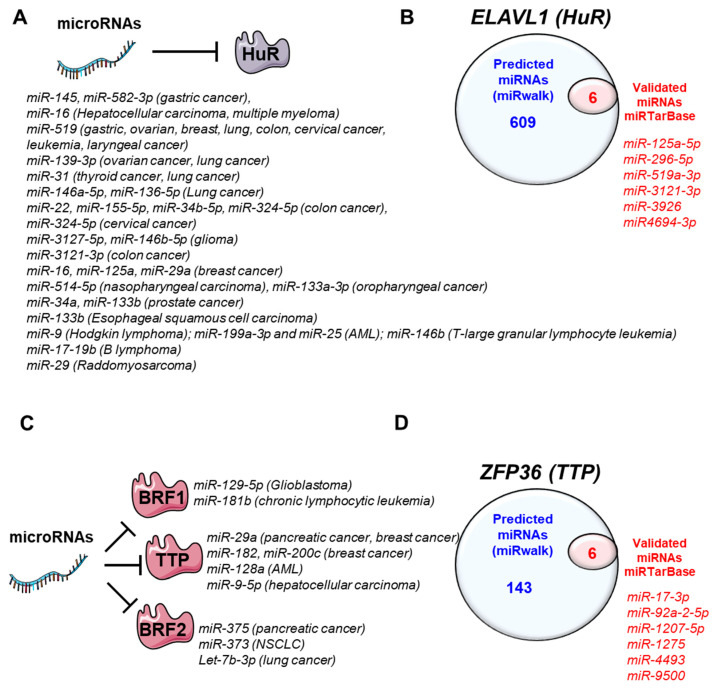
MicroRNAs regulating HuR and TTP family members. (**A**) miRNAs regulating HuR expression and frequently downregulated in cancers. (**B**) Predicted and validated miRNAs regulating HuR (miRWalk database: http://mirwalk.umm.uni-heidelberg.de/, accessed on 13 June 2022). (**C**) miRNAs regulating TTP family members (TTP, BRF1 and BRF2) expression and frequently upregulated in cancers. (**D**) Predicted and validated miRNAs regulating TTP expression in cancers (miRWalk database: http://mirwalk.umm.uni-heidelberg.de/, accessed on 13 June 2022).

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
