# Peer review of "MicroRNAs, Tristetraprolin Family Members and HuR: A Complex Interplay Controlling Cancer-Related Processes"

_cancers, 2022, doi:10.3390/cancers14143516_

Round 1

Reviewer 1 Report

The article by Cyril Sobolewski and colleagues entitled “MicroRNAs and AU-Rich Element Binding Proteins: A poorly considered interplay with potential therapeutic applications in cancers” is a simple review. The authors try to summarize the important interplay of microRNAs and AU-rich element binding proteins. Altogether, this review is on a low level and important work in this field have not been cited. Therefore, this review cannot be published in the journal “cancers”.

My major concerns are:

- All abbreviations must be introduced before they are used.

- Very often citations for statements are missing; the authors have to prove all statements with citations. There are so many missing citations that I cannot mention all here.

- If the authors state something like “…most studies….” more than one citation is necessary.

- AUBP must be added to the keywords.

- The authors must be more accurate and statements like etc are not scientific at all.

- Table 1 is missing.

- The authors have to explain why they use sometimes bold print in the text body.

- The chapter “Take home message” is not nice at all. Beside the fact that once again citations are missing (like in all other parts of the manuscript). The authors have to fuse these points with the “Conclusion” part. By the way the “Conclusion” part is weak and must be improved significantly.

Author Response

The article by Cyril Sobolewski and colleagues entitled “MicroRNAs and AU-Rich Element Binding Proteins: A poorly considered interplay with potential therapeutic applications in cancers” is a simple review. The authors try to summarize the important interplay of microRNAs and AU-rich element binding proteins. Altogether, this review is on a low level and important work in this field have not been cited. Therefore, this review cannot be published in the journal “cancers”.

We thank the reviewer for their time and for their constructive comments. Following the comment of the reviewer, we screened again literature by using different key words and we could identify additional studies that were indeed not discussed in our manuscript. We apologize for this mistake and all these references are now discussed and cited in the revised version. We also added several references associated to the general information of the manuscript (e.g., presentation of TTP family, HuR proteins etc). Together, more than 50 references have been added in our revised manuscript.

We would like to precise that the goal of our review is to sensitize the readership to the importance of the interplay between miRNAs and AUBPs in cancer-related processes, in line with the topic of this special issue. Moreover, we decided to emphasize on tristetraprolin family members (TTP, BRF1 and BRF2) and HuR, which are the most studied AUBPs in cancers. We therefore did not discuss extensively the interplay between miRNAs and all the other AUBPs. We believe that documenting all these other interactions will render the review poorly comprehensive for the readership. To avoid misunderstanding and to be more consistent with the content of our manuscript, we changed our title as follow: “MicroRNAs, tristetraprolin family members and HuR: A complex interplay controlling cancer-related-processes”.

Besides, we are aware that several other connections between miRNAs, lncRNA and AUBPs exist but have not necessarily been described in a cancerous context. Indeed, many studies have reported an interplay between miRNAs and AUBPs in other contexts (e.g., cardiovascular disease, inflammatory disorders, metabolic diseases, viral infection), which are out of the scope of our review on carcinogenesis. We also excluded studies dealing with other networks, independently of microRNAs. However, we agree with the reviewer that these other links, are also highly relevant for cancer development. We therefore opened our conclusion to these other interplay in our new conclusion. We hope that the reviewer will appreciate our efforts to improve our manuscript.

My major concerns are:

- All abbreviations must be introduced before they are used.

We carefully checked and explained all the abbreviations of our manuscript.

- Very often citations for statements are missing; the authors have to prove all statements with citations. There are so many missing citations that I cannot mention all here.

We carefully reviewed our manuscript and we have added missing references when it was necessary.

- If the authors state something like “…most studies….” more than one citation is necessary.

We have added more citations as requested by the reviewer.

- AUBP must be added to the keywords.

AUBP” has been added in the keywords.

- The authors must be more accurate and statements like etc are not scientific at all.

All “etc” have been removed.

- Table 1 is missing.

We thank the reviewer for noticing this mistake. Table 1 was reported as “table 2”.  We have corrected this issue, and we have included the reference of table 1 in the body text.

- The authors have to explain why they use sometimes bold print in the text body.

This issue was generated when the MS template was uploaded on the Cancers website during the submission process. We have corrected these issues and we carefully checked again in the whole manuscript.

- The chapter “Take home message” is not nice at all. Beside the fact that once again citations are missing (like in all other parts of the manuscript). The authors have to fuse these points with the “Conclusion” part. By the way the “Conclusion” part is weak and must be improved significantly.

We changed our conclusion to better highlight the interest of our review and the key concept that should be underlined. As suggested by the reviewer, the “Take home message” paragraphs have been incorporated in our new conclusion.

Reviewer 2 Report

The manuscript provides a review of the relationship between AUBP and miRNAs, especially the three main classes of interaction mechanisms in cancer. It is a complete and informative review, but does not give much valuable contribution to the importance of the field and the future direction of development. The shortcomings are as follows.

  1. the quality of the figures in the manuscript needs to be improved, with many text and graphics being obscured from each other
  2. the format of the article is not uniform, some paragraphs are bolded, such as HuR-circ-CCND1-miR-646, HuR-LIN00336-miR6852
  3. The manuscript title emphasizes therapeutic applications, but the article content does not provide relevant cancer treatment strategies and application prospect introduction or reflection

Author Response

The manuscript provides a review of the relationship between AUBP and miRNAs, especially the three main classes of interaction mechanisms in cancer. It is a complete and informative review but does not give much valuable contribution to the importance of the field and the future direction of development. The shortcomings are as follows.

We thank the reviewer for their time to review our manuscript. We believe that our review provides key information to better understand the post-transcriptional regulation of cancer-related genes, which is an important prerequisite for the development of novel therapeutic approaches. Our review highlights the importance of considering the interplay between miRNA and AUBPs, rather than their individual functions. By giving several examples, this review shows that this interplay is critical for miRNA activity, thus questioning the relevance of studies assessing miRNA levels only. Furthermore, this interplay questions the dogmatic view of some miRNAs (e.g., miR-21) or AUBPs (e.g., HuR) as strict oncogenes or tumor suppressors and thus question also the potential therapeutic approaches, aiming at targeting them (e.g., HuR inhibitors, Anti-miR-21). As explained in our review, this interplay is poorly considered in the field of post-transcriptional gene regulation and thus our current comprehension of miRNA biology in physiological and pathological processes is strongly limited. Moreover, very few studies and reviews are considering this interplay, in particular, in the context of cancer. We therefore believe that our efforts to integrate this complex network in the setting of carcinogenesis represent a valuable contribution to the field. We believe that deciphering the complexity of this regulatory network is a major challenge for precision medicine because the different links discussed in our review are associated with the prognosis of the patient, chemosensitivity and provide a comprehensive view of active/inactive miRNAs prior to any therapeutic interventions. Although, most of these points are discussed in the introduction and the core of the review, we changed our conclusion to better highlight the importance of these concepts and to provide future directions:

  • The use of in vivo models to study the relevance of this interplay in cancers in a more physiological system.
  • Combining different alterations in our models to be closer to the human pathology (e.g., HuR and miR-21 overexpression + TTP loss).
  • Deciphering miRNAs regulating oncogenic/tumor suppressive AUBPs for the design of new therapeutic approach.
  • Developing new bioinformatic tools (database repository, deep learning, artificial intelligence) to model properly this interplay for each cancer. This is a key step for precision medicine. This modeling has previously been suggested in the following study, which is now cited in our review (PMID: 26837572)

We changed also figure 1 to better highlight the importance of these networks in the regulation of oncogenes and tumor suppressors. For a better clarity, we have separated figure 1 in two different figures.

Finally, we considered all the comments of the reviewer mentioned above and we hope that the reviewer will appreciate our efforts to improve our review.

1. the quality of the figures in the manuscript needs to be improved, with many text and graphics being obscured from each other

We apologize for the issue encountered by the reviewer, which could be due to the pdf conversion during our submission. However, this is not the case in our word file. We separated the figure 1 in two different figures to improve the clarity. We will carefully check this issue during the next steps of our submission and the raw files of the figures will be provided to the editorial office.

2. the format of the article is not uniform, some paragraphs are bolded, such as HuR-circ-CCND1-miR-646, HuR-LIN00336-miR6852

We apologize for this problem, which was again generated during the submission process of our template. All these issues have been corrected.

3. The manuscript title emphasizes therapeutic applications, but the article content does not provide relevant cancer treatment strategies and application prospect introduction or reflection

We agree with the reviewer that the therapeutic approaches are not really discussed, and we apologize for this misunderstanding. Our review is mostly highlighting the importance of the interplay between miRNAs and AUBPs in cancer-related processes. Some of these interplays suggest a dual function of some miRNAs/AUBPs in carcinogenesis, thus raising question regarding their therapeutic targeting in cancer. We therefore changed the title of our review as follow: “MicroRNAs, tristetraprolin family members and HuR: A complex interplay controlling cancer-related-processes”.

Round 2

Reviewer 1 Report

The authors have improved the manuscript and the current version can be published.